# Vinylene-Linked Emissive Covalent Organic Frameworks for White-Light-Emitting Diodes

**DOI:** 10.3390/polym15183704

**Published:** 2023-09-08

**Authors:** Yan Li, Xiaohan Wu, Jinyi Zhang, Congcong Han, Mengmeng Cao, Xiangrong Li, Jieqiong Wan

**Affiliations:** College of Chemistry and Chemical Engineering, Shanghai University of Engineering Science, Shanghai 201620, China; liyan199603@163.com (Y.L.);

**Keywords:** covalent organic frameworks, emissive, photoluminescence, stability, WLEDs

## Abstract

Covalent organic frameworks (COFs) have gained considerable attention due to their highly conjugated π-skeletons, rendering them promising candidates for the design of light-emitting materials. In this study, we present two vinylene-linked COFs, namely, VL-COF-1 and VL-COF-2, which were synthesized through the Knoevenagel condensation of 2,4,6-trimethyl-1,3,5-triazine with terephthalaldehyde or 4,4′-biphenyldicarboxaldehyde. Both VL-COF-1 and VL-COF-2 exhibited excellent chemical and thermal stability. The presence of vinylene linkages between the constituent building blocks in these COFs resulted in broad excitation and emission properties. Remarkably, the designed VL-COFs demonstrated bright emission, fast fluorescence decay, and high stability, making them highly attractive for optoelectronic applications. To assess the potential of these VL-COFs in practical devices, we fabricated white-light-emitting diodes (WLEDs) coated with VL-COF-1 and VL-COF-2. Notably, the WLEDs coated with VL-COF-1 achieved high-quality white light emission, closely approximating standard white light. The promising performance of VL-COF-coated WLEDs suggests the feasibility of utilizing COF materials for stable and efficient lighting applications.

## 1. Introduction

In recent decades, the solid-state lighting industry has witnessed a significant transformation with the introduction of phosphor-converted white-light-emitting diodes (pc-WLEDs). These devices have gained immense popularity due to their exceptional energy efficiency, long-lasting nature, and remarkable versatility [1,2]. In pc-WLEDs, the phosphor that is utilized plays a crucial role in determining the overall performance of the devices. Therefore, it becomes imperative to explore and utilize novel phosphors that exhibit strong luminescence, high quantum efficiency, and chemical stability. These characteristics are essential in the development of high-performance pc-WLED devices [3,4,5]. The typical commercial WLEDs are achieved by coating a rare-earth-doped yttrium aluminum garnet yellow phosphor onto the surface of a blue gallium nitride chip [6,7]. Recently, the domain of phosphors for WLED lighting has witnessed an emergence of and subsequent surge in interest surrounding three intriguing contenders: halide perovskites, metal–organic frameworks (MOFs), and covalent organic frameworks (COFs). These novel materials have garnered substantial attention within the research community, owing to their potential as highly efficient and versatile phosphors [8,9,10,11]. In the field of rare-earth-free phosphors for WLEDs, emissive polymers, particularly emissive COFs, have emerged as a promising and cutting-edge frontier. The significant progress in this area began with the pioneering work that was published in 2018 [12]. COFs represent a unique class of crystalline polymers distinguished by their interconnected framework structure, which is solely based on strong covalent bonds. This distinctive feature sets COFs apart and renders them highly intriguing for applications in WLED technology. These COFs exhibit a systematic arrangement of nano-sized pores and channels, providing them with unique properties [13,14,15]. The integration of light elements such as C, H, O, B, and N facilitates the formation of robust organic bonds, contributing to the exceptional chemical stability and low density exhibited by COFs [16,17,18]. Furthermore, the synthesis of COFs involves reversible organic reactions, which not only enable the construction of the desired framework, but also facilitate self-healing mechanisms to rectify any defective fragments. As a consequence, COFs possess exceptional crystallinity and porosity, making them highly attractive for diverse applications [19,20,21].

Initially recognized for their exceptional surface area, gas adsorption capacity, and diverse applications in catalysis and sensing, COFs have witnessed an exciting transformation by venturing into the realm of light-emitting materials. One of the distinguishing features of COFs is their ability to exhibit tunable emission properties, making them highly attractive for the design of next-generation optoelectronic devices. COFs have now emerged as a fascinating avenue for the development of novel light-emitting materials, opening up new possibilities for their utilization in various technological applications. [18,22,23,24,25,26,27,28]. The controlled incorporation of ordered π-conjugated building blocks or functional groups into specific COFs allows for the efficient migration of excitons and facilitates energy transfer, making them promising candidates for emission centers. This potential enables the design and fabrication of resilient emissive COF phosphors. Their remarkable potential also arises from the synergistic combination of inherent porosity and the ability to incorporate luminescent moieties within their crystalline networks. These new luminescent COFs represent a compelling bridge between traditional organic emissive materials and well-established porous frameworks. As a result, emissive COFs exhibit remarkable characteristics, such as the ability to adjust emission wavelengths, achieve high quantum yields, and maintain exceptional stability. These outstanding attributes position emissive COFs as highly promising contenders for a wide range of applications in the fields of optoelectronics and sensing [12,29,30,31,32,33,34,35,36,37]. In a pioneering study conducted by Wang et al. in 2018, emissive COFs were successfully utilized in the development of WLEDs. The fabricated WLEDs employed a 3D-TPE-COF, along with a blue-emitting chip. Notably, these COF-based WLEDs exhibited exceptional stability when subjected to continuous operation under ambient conditions for an impressive duration of 1200 h, without any observable degradation. This groundbreaking achievement highlights the promising potential of emissive COFs as reliable and long-lasting materials for the advancement of lighting technology [12]. Developing fluorescent COF materials has indeed posed significant challenges due to their structural rigidity, susceptibility to π-conjugation breaks, and quenching effects. As a result, the research on COF phosphors has been limited. However, notable contributions have been made in this field, demonstrating promising avenues for further investigation. For instance, Huang et al. designed two emissive sp^2^-C-COFs: TAT-COF and TPB-COF. They successfully achieved white light emission by physically mixing the blue-emitting TAT-COF with the yellow-emitting TPB-COF. Their study highlighted the importance of several key factors, including planarity, conjugation, orientation of the dipole moment, and interlayer aggregation, in controlling the light-harvesting ability of COFs and the exciton relaxation pathway. This work provided insights which are valuable for enhancing the photoluminescent quantum yield of COFs [35]. Wang et al. reported the synthesis of a sp^2^-carbon-linked COF that exhibited emission at a peak wavelength of 582 nm. They further demonstrated the fabrication of a series of cold WLEDs by adjusting the number of COFs. The yellow-emitting COFs developed in their study exhibited a high photoluminescent quantum yield (PLQY) and good thermal stability, making them suitable for use as metal-free phosphors in LEDs [38]. In another study, Jiao et al. presented a fully π-conjugated 2D COF comprising triazine units. This COF exhibited emission at a peak wavelength of 581 nm. The researchers successfully fabricated various WLEDs, including cold, neutral, and warm variants, based on this material. Their work showcased the potential of π-conjugated COFs to achieve tunable emission properties for different lighting applications [39]. Wang et al. synthesized imine- and vinylene-linked pyrene-based COFs, which exhibited vastly different solid-state PLQY attributed to the distinct bonding modes employed. The imine-based COF displayed a PLQY of 0.34%, whereas the vinylene-linked COF exhibited a significantly higher PLQY of 15.43%. By coating the vinylene-linked COF onto a LED strip, they successfully developed an effective white light-emitting device. Furthermore, the researchers elucidated the influence of different charge transfer pathways in imine- and vinyl-linked COFs on the exciton relaxation way and fluorescence intensity. These findings provide valuable insight into the manipulation of the excitation-energy transfer process and emission properties in COF materials [40]. These studies demonstrate significant progress in the development of COF phosphors. By exploring different COF structures, optimizing synthesis strategies, and investigating the effects of various factors on fluorescence properties, researchers are advancing our understanding and capabilities in this field. Continued research on COFs holds promise for the future development of efficient and customizable fluorescent materials.

In this work, two vinylene-linked COFs, namely, VL-COF-1 and VL-COF-2, were successfully synthesized through Knoevenagel condensation by condensing 2,4,6-trimethyl-1,3,5-triazine with either terephthalaldehyde or 4,4′-biphenyldicarboxaldehyde (see Figure 1). The synthesis of sp^2^ carbon-conjugated frameworks in VL-COF-1 and VL-COF-2 was effectively achieved, offering numerous benefits such as extended π-conjugation resulting from increased C=C bonding and exceptional crystallinity [41]. The fully conjugated backbone of VL-COF-1 and VL-COF-2 enables uninterrupted π-electron delocalization, thereby facilitating efficient migration and transport of photons, holes, and electrons. This characteristic makes them highly appealing for their luminescent properties. By employing this strategy, the stability and luminescent performance of COFs have been significantly enhanced [42]. In order to investigate the formation of π–π stacks in both VL-COFs, Fourier transform infrared spectroscopy (FTIR) was utilized. The FTIR spectra, as depicted in Appendix A, exhibited distinct stretching vibration bands at 1630 cm^−1^ and 974 cm^−1^ for VL-COF-1 and VL-COF-2, respectively. These bands are indicative of the successful formation of C=C bonds within the COFs. It is noteworthy that these C=C bonds were introduced through the incorporation of vinylene linkages between the pristine building blocks, as illustrated in Figure 1. Furthermore, the decrease in intensity observed in the C=O stretching vibration peak (1690 cm^–1^) of terephthalaldehyde or 4,4′-biphenyldicarboxaldehyde monomers provides supplementary evidence in favor of a pronounced degree of polymerization within the COF structure [43,44]. In addition, solid-state nuclear magnetic resonance (NMR) measurements were performed, and the corresponding data are presented in Appendix A. The distribution of peaks observed in Appendix A confirms the formation of C=C bonds in compounds VL-COF-1 and VL-COF-2, providing further evidence of the synthesis of the target product. This observation aligns with the results reported in the relevant literature [45,46]. However, due to the typically lower resolution, analysis of the ^1^H MAS solid-state NMR spectra in Appendix A is often deemed unreliable. Therefore, further interpretation of Appendix A was not pursued. The prepared VL-COF-1 and VL-COF-2 were yellow solid powders that were insoluble in common organic solvents and water; hence, liquid-NMR spectroscopy was not used for testing. The electronic absorption spectra of the two VL-COFs were examined through ultraviolet-visible absorption spectroscopy analysis, using the monomers as reference points. The obtained data, presented in Appendix A, clearly showed a significant red shift in the absorption peaks of the VL-COF samples compared to the building units in the pristine monomers. This observation suggests an expansion of the π-conjugation within the frameworks, indicative of enhanced electron delocalization. The observed red shift in the absorption spectra of the VL-COFs provides valuable insight into the structural modifications and electronic properties of these materials, highlighting their potential for various applications in the field of optoelectronics. Thermogravimetric analysis (TGA) was conducted in a nitrogen environment to evaluate the thermal stability of the two VL-COFs, as illustrated in Appendix A. Both COFs exhibited excellent thermal stability and retained their structural integrity even at high temperatures, specifically up to 400 °C. The thermal stability of the COFs was assessed through rigorous analysis, demonstrating their capabilities to endure thermal conditions. To further understand their structural characteristics, field-emission scanning electron microscopy (FE-SEM) was employed, revealing the unique stacking morphologies of the VL-COFs.

The porous characteristics of the two VL-COFs were investigated through the measurements of nitrogen adsorption–desorption at a temperature of 77 K. Both VL-COF-1 and VL-COF-2 exhibited a type-I sorption curve, indicating the presence of a micropore structure (see Appendix A). To determine the specific surface area of the VL-COF materials, the Brunauer–Emmett–Teller (BET) method was employed, resulting in values of 485 and 317 m^2^ g^−1^ for VL-COF-1 and VL-COF-2, respectively. Moreover, the pore volumes of these materials were calculated based on the nitrogen adsorption curve at P/P_0_ = 0.99, yielding values of 0.51 and 0.29 cm^3^ g^−1^ for VL-COF-1 and VL-COF-2, respectively. Further characterization of the pore sizes of VL-COF-1 and VL-COF-2 was conducted using the non-local density functional theory (NLDFT) method. The NLDFT analysis yielded pore sizes of approximately 1.2 and 1.8 nm for VL-COF-1 and VL-COF-2, respectively, as illustrated in Appendix A. The presence of these nanometer-sized pores holds great significance as it is expected to enhance the light output of the COFs, thus offering potential benefits in various relevant applications.

## 2. Materials and Methods

Materials: Dioxane, mesitylene, acetic acid (HOAc), terephthalaldehyde, and 4,4′-biphenyldicarboxaldehyde were procured from reputable suppliers, including TCI (Tixiai (Shanghai) Chemical Industry Development Co., Ltd., Shanghai, China), Sigma-Aldrich (Sigma Aldrich Shanghai Trading Co., Ltd., Shanghai, China), and Wako (FUJIFILM Wako (Guangzhou) Trading Co., Guangzhou, China). The commercial optical epoxy resin used in the study was obtained from Struers (Struers Ltd., Ballerup, Denmark).

Preparation of VL-COF-1: 2,4,6-Trimethyl-1,3,5-triazine (24.6 mg, 0.20 mmol), terepthalaldehyde (40.2 mg, 0.30 mmol), dioxane (1.0 mL), mesitylene (1.0 mL), acetonitrile (0.1 mL), and trifluoroacetic acid (0.5 mL) were added into the system. The tube was then flash-frozen at 77 K and degassed by three freeze–pump–thaw cycles. The tube was sealed off and then heated at 150 °C for 3 days. The collected powder was washed with N,N′-dimethylacetamide, tetrahydrofuran, ammonia solution, methanol, and acetone several times, soxhleted by tetrahydrofuran for 5 h, and then dried at 100 °C under vacuum for 12 h to obtain the VL-COF-1 sample (Yield: 89%, 49.1 mg, relative to the amount of monomer used).

Preparation of VL-COF-2: 2,4,6-Trimethyl-1,3,5-triazine (24.6 mg, 0.20 mmol), biphenyldicarboxaldehyde (63.0 mg, 0.30 mmol), dioxane (1.0 mL), mesitylene (1.0 mL), acetonitrile (0.1 mL), and trifluoroacetic acid (0.5 mL) were added into the system. The tube was then flash-frozen at 77 K and degassed by three freeze–pump–thaw cycles. The tube was sealed off and then heated at 150 °C for 3 days. The collected powder was washed with N,N′-dimethylacetamide, tetrahydrofuran, ammonia solution, methanol, and acetone several times; soxhleted by tetrahydrofuran for 5 h; and then dried at 100 °C under vacuum for 12 h to obtain the VL-COF-2 sample (Yield: 85%, 66.3 mg, relative to the used amount of monomer).

Fabrication of COF-coated WLEDs: For the preparation of COF-coated WLEDs, we introduced VL-COFs (0.1 g) into a mixture comprising component A (1 g) and component B (0.5 g) of an optical epoxy resin, SpeciFix Resin, along with SpeciFix-40 Curing Agent. This combination was carefully formulated at a weight ratio of 0.5:5:2.5. The resulting mixture underwent thorough stirring until a uniform and homogeneous slurry was obtained. The slurry was then applied in a consistent and even manner onto the surface of a commercially available blue LED, which emitted light at a peak wavelength of approximately 450 nm. The coating process implemented in this study can be easily reproduced, ensuring a continuous and uniform deposition of COFs over the surface of the blue LED chip.

Characterizations: Infrared spectra were acquired using an Avatar FT-IR 360 spectrometer (Thermo Fisher, Waltham, MA, USA) over the spectral range of 600 to 3500 cm^−1^. The ^1^H magic angle spinning (MAS) solid-state NMR spectrum was obtained using a Bruker Avance III HD 400 spectrometer (Bruker Corporation, Rheinstetten, Germany). The experimental setup involved a magnetic field strength of 9.3947 T, a resonance frequency of 79.49 MHz, and a spinning speed of 10 kHz. In addition, the ^13^C cross-polarization (CP)/MAS solid-state NMR spectra were acquired using the same Bruker spectrometer. The measurements entailed a magnetic field intensity of 9.3947 T, a resonance frequency of 100.61 MHz, a spinning speed of 5 kHz, and a 2 ms CP pulse duration. Powder X-ray diffraction data were acquired using a PANalytical BV Empyrean diffractometer (Malvern Panalytical Ltd., Malvern, UK) by applying powder samples onto a glass substrate. The diffraction measurements were performed over a 2θ range of 4.0° to 30° with a step size of 0.02°. Thermogravimetric analysis (TGA) was conducted using a TA Q500 thermogravimeter (TA Instruments, New Castle, DE, USA) in a controlled nitrogen atmosphere. The temperature was raised from room temperature to 800 °C at a heating rate of 10 °C min^−1^. Nitrogen sorption isotherms were measured at a temperature of 77 K using a JW-BK 132F analyzer (Beijing JWGB Sci & Tech Co., Beijing, China). Field emission scanning electron microscopy and energy dispersive spectroscopy for elemental mapping were recorded on a JSM-7001F microscope (JEOL Ltd., Tokyo, Japan). This experimental setup allowed for the acquisition of crucial data on the spatial distribution and elemental composition within the sample. The quantum efficiency was determined using the FLS1000 (Edinburgh Instruments Ltd., Livingston, UK) quantum efficiency measurement system, employing the integrating sphere method (the direct method). The photoluminescence (PL) decay curves were acquired by utilizing an FLS1000 fluorescence spectrometer equipped with an nF900 nanosecond flash lamp as the excitation source. The photoluminescence (PL) and photoluminescence excitation (PLE) spectra were acquired by means of the Hitachi F-4700 Fluorescence Spectrophotometer (Hitachi, Ltd., Tokyo, Japan) with a Xe lamp. To investigate the electron luminescence (EL) characteristics of the WLED lamp, an HP350C spectral colorimeter (Hangzhou Hopoo Light & Color Technology Co., Ltd., Hangzhou, China) was used to measure and record the relevant properties. This specialized equipment enabled the collection of photometric and colorimetric data regarding the EL emission from the WLED lamp.

## 3. Results and Discussion

The crystallinity of the two newly developed VL-COFs was evaluated through powder X-ray diffraction (PXRD) analysis, as depicted in Figure 2a,c. For VL-COF-1, strong peaks at 4.80°, 8.32°, 10.54°, 15.64°, and 26.64° (Figure 2a) were identified, corresponding to the 100, 110, 200, 300, and 001 facets, respectively. VL-COF-2 exhibited signals at 3.48°, 6.04°, 9.14°, and 14.42° (Figure 2c, black), corresponding to the 100, 110, 200, and 300 facets, respectively. The experimental diffraction patterns were assigned using the Pawley refined pattern, as seen in Figure 2a,c (black lines). Based on the PXRD patterns, among various possible stacking modes, the AA stacking mode was found to be in good agreement with the observed patterns of both VL-COF-1 and VL-COF-2 (Figure 2a,c, green). The unit cells corresponding to AA stacking are further illustrated in Figure 2b,d, providing a visual representation of the atomic arrangement in these materials. In contrast, the AB staggered stacking mode did not adequately reproduce the observed PXRD experimental results (Figure 2a, purple). The deviations between the predicted and observed diffraction signals indicate that the AB stacking arrangement was not in agreement with the crystal structures of VL-COF-1 and VL-COF-2. The lattice parameters of VL-COF-1 were optimized as follows: a = b = 21.3258 Å, c = 3.3776 Å, *α* = *β* = 90°, and *γ* = 120°, with error factors of R_wp_ = 1.96% and R_p_ = 1.58% (blue curve in Figure 2a). In a similar manner, the optimized lattice parameters for VL-COF-2 were determined as: a = b = 29.8423 Å, c = 3.3996 Å, *α* = *β* = 90°, and *γ* = 120°, with R_wp_ of 5.93% and R_p_ of 4.95% (blue curve in Figure 2c).

To evaluate the chemical stability of VL-COF-1 and VL-COF-2, the COF powders were immersed in various solvents under specific conditions, including distilled water, N,N-dimethylformamide (DMF), and aqueous solutions of HCl (6 M) and NaOH (6 M). The immersion process was carried out at room temperature for a duration of 72 h. After the immersion process, the VL-COF powders were collected with the utmost care, and subsequent FTIR and XRD analyses were conducted. The obtained FTIR spectra exhibited vibration bands that closely resembled those observed in the original as-synthesized COFs. Appendix A present the FTIR spectra of the VL-COF-1 and VL-COF-2 samples, respectively, highlighting the similarity in vibration bands with the as-synthesized COFs. This observation suggests that their chemical structure was effectively preserved. Additionally, the primary strong PXRD patterns of VL-COF-1 and VL-COF-2 remained unaffected after the immersion process, further validating their exceptional chemical stability (see Appendix A). Through these analytical techniques, it was confirmed that the prepared VL-COFs exhibited remarkable resistance to chemical degradation. Despite prolonged exposure to different solvents, the structures of VL-COF-1 and VL-COF-2 remained intact. These findings highlight the potential suitability of VL-COFs for diverse applications that necessitate robust chemical stability.

Following the confirmation of their structural features and chemical stability, the emission activity of VL-COF-1 and VL-COF-2 was thoroughly investigated. The emission characteristics of VL-COF-1 were assessed, revealing a broad excitation spectrum spanning a range of 350–480 nm, with a peak at 430 nm (as shown in Figure 3a). This excitation profile aligned well with the emission spectrum of blue light-emitting diodes, which is advantageous for potential optoelectronic applications. The emission spectrum of VL-COF-1 extended from 500 to 700 nm, exhibiting a broad band centered at 550 nm, indicative of typical yellow emission. Similarly, VL-COF-2 also demonstrated comparable excitation and emission properties (as depicted in Figure 3b). These results highlight the potential of both VL-COF-1 and VL-COF-2 as promising materials for diverse applications in the field of light emission.

Among them, compared with VL-COF-1, the maximum emission wavelength of VL-COF-2 (λ_em_ = 525 nm) is redshifted because of a certain spatial rotation in the biphenyl structure of the target product, which weakened the conjugation degree of VL-COF-2 to some extent, consequently impacting its luminescence performance. These findings provide strategies for designing COFs with specific luminescence properties. The absolute fluorescence quantum yield of the VL-COF-1 and VL-COF-2 powder was measured to be 12.7% and 11.2%, respectively, using the integrating sphere method. It is precisely the intramolecular rotation of the biphenyl structure in VL-COF-2 that leads to the nonradiative decay of the excited state, resulting in a lower quantum efficiency compared to VL-COF-1. To compare the luminescent properties of the COFs reported in this study with previously reported COFs, Appendix A was compiled. The COFs synthesized in this work exhibited a moderate level of quantum efficiency. Both COFs demonstrated fluorescence decay, with fluorescence lifetimes of 10.12 and 7.51 ns, respectively. Notably, VL-COF-1 exhibited a longer fluorescence lifetime, suggesting a higher degree of π-electron delocalization and framework conjugation. This characteristic facilitates efficient migration of photons, holes, and electrons, making VL-COF-1 more favorable for optoelectronic applications.

In order to capitalize on the yellow emission of VL-COFs and to achieve white light emission, we devised a method to fabricate COF-coated white light-emitting diode (WLED) devices. This involved uniformly dispersing VL-COF-1 and VL-COF-2 in epoxy resin and subsequently coating the mixture onto commercially available blue LED chips. The detailed procedure can be found in the fabrication of COF-coated WLEDs. As shown in Figure 4b,d, the composite of yellow COFs and epoxy resin covered the surface of the LED chip in a cylindrical shape. Upon turning on the light, bright white light was observed. The electroluminescent spectra of the two WLEDs, fabricated using VL-COF-1 and VL-COF-2, were characterized, and the resulting white light was analyzed for Commission Internationale de l’Eclairage (CIE) coordinates, color rendering index (CRI), and correlated color temperature (CCT). The corresponding data are illustrated in Figure 4a,c. The chromaticity diagram containing the CIE coordinates of the blue LED, VL-COF-1, and VL-COF-1 and the white light is illustrated in Figure 4e. VL-COF-1 demonstrated improved white light quality due to its higher red-light component, as evidenced by its CIE coordinate of (0.33, 0.34). This coordinate closely approached the standard coordinates for white light (0.33, 0.33). The CCT values of the two fabricated WLEDs were measured at 5337 K and 7656 K. The former fell within the range of neutral white light, closely resembling the color temperature of conventional tubular fluorescent lamps. Conversely, the latter exhibited cool white light characteristics. Notably, the WLED device incorporating VL-COF-1 achieved a CRI of 79.1, surpassing the CRI of a commercial WLED light source which utilized a yellow-emitting Y_3_Al_5_O_12_:Ce^3+^ phosphor and a blue InGaN chip [47,48]. This indicates that the fabricated WLED device can offer superior light quality. The results presented here highlight the significant potential of VL-COF-1 as a rare-earth-free phosphor for WLED lighting applications.

## 4. Conclusions

In conclusion, the newly synthesized VL-COFs, specifically VL-COF-1 and VL-COF-2, exhibited impressive characteristics that position them as promising candidates for light-emitting materials. These COFs displayed wide emission spectra, rapid fluorescence decay, and exceptional stability. Notably, when combined with blue light-emitting diodes, they demonstrated the potential to generate white light efficiently. The resultant COF-coated WLED devices delivered bright white light emission, exhibiting high color rendering indices (Ra = 79.1) and correlated color temperatures (CCT = 5337 K). Consequently, these COFs can serve as environmentally friendly alternatives to rare-earth-based phosphors, facilitating the development of efficient and sustainable solid-state lighting technologies. These findings highlight the significance of emissive COFs as cutting-edge phosphor materials, paving the way for sustainable and high-performance solid-state lighting technologies.

## Figures and Tables

**Figure 1 polymers-15-03704-f001:**
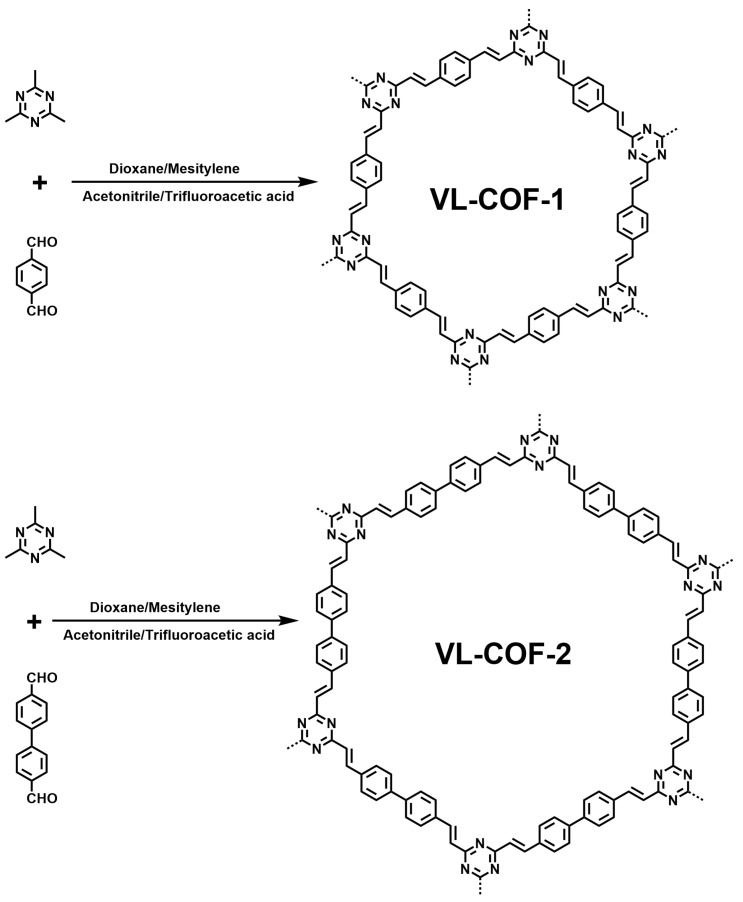
Design and synthesis of VL-COFs.

**Figure 2 polymers-15-03704-f002:**
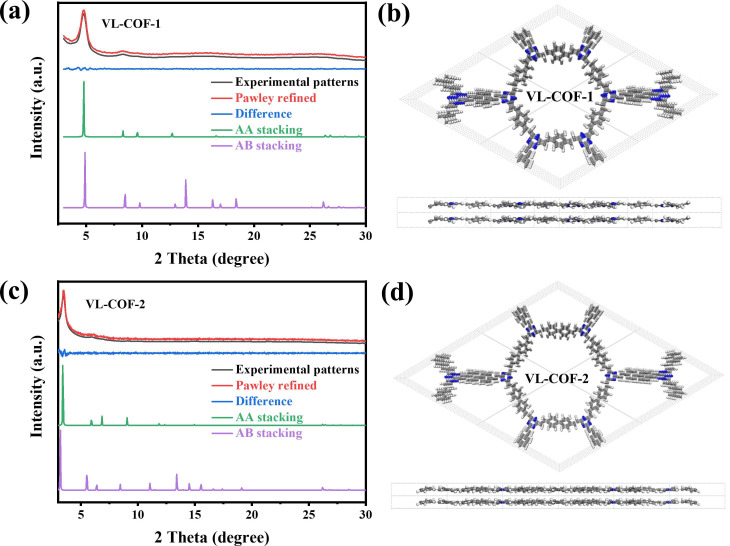
PXRD patterns of (**a**) VL-COF-1 and (**c**) VL-COF-2. (Experimental patterns: black; Pawley refined: red; their difference: blue; staggered AA-stacking mode: green; staggered AB-stacking mode: purple). Unit cells of AA-stacking for (**b**) VL-COF-1 and (**d**) VL-COF-2.

**Figure 3 polymers-15-03704-f003:**
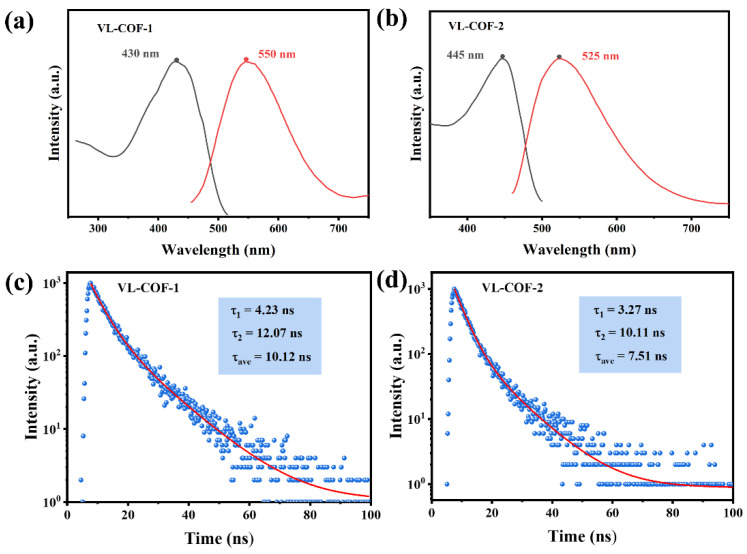
Normalized fluorescence excitation (black) and emission (red) spectra of (**a**) VL-COF-1 (the monitoring wavelength is λ = 550 nm and the excitation wavelength is λ = 430 nm) and (**b**) VL-COF-2 (the monitoring wavelength is λ = 525 nm and the excitation wavelength is λ = 445 nm). Fluorescence decay curves of (**c**) VL-COF-1 and (**d**) VL-COF-2.

**Figure 4 polymers-15-03704-f004:**
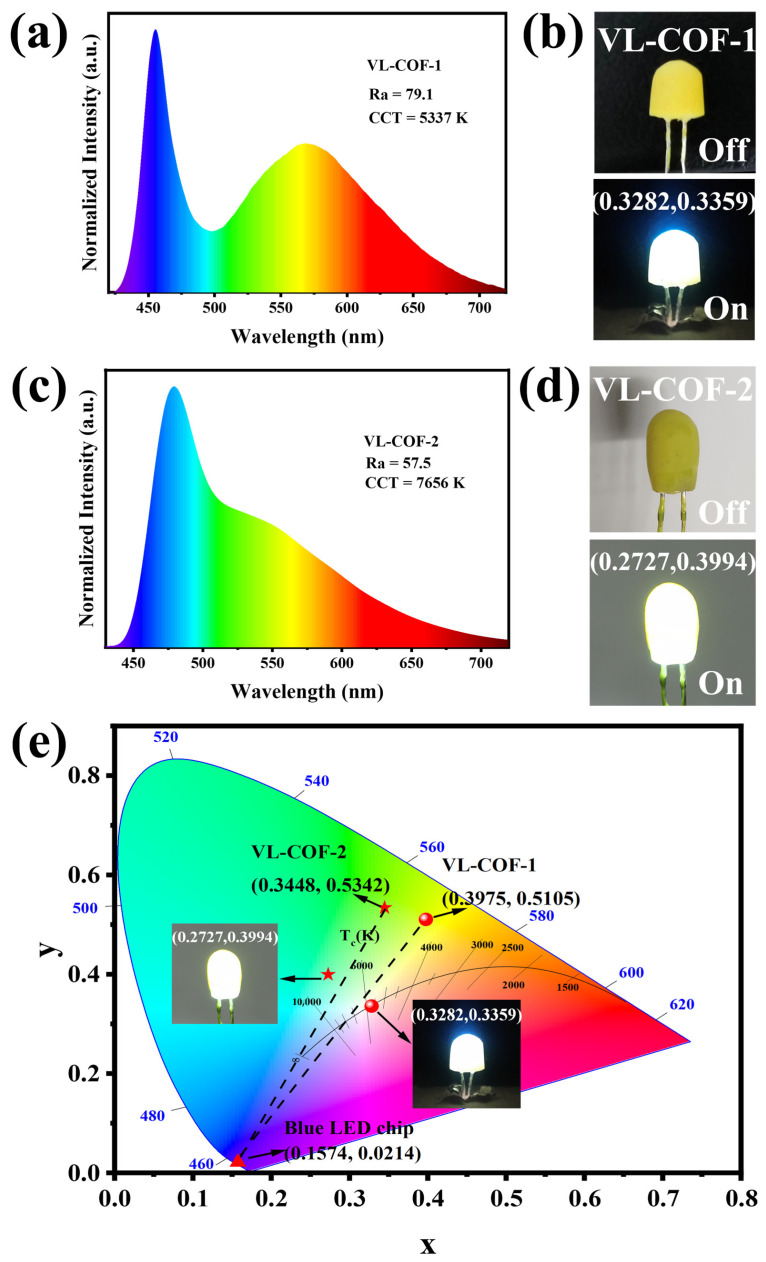
The performance of the VL-COF-coated WLEDs. Electroluminescence (EL) spectra of (**a**) VL-COF-1- and (**c**) VL-COF-2-coated WLEDs. Digital photos of (**b**) VL-COF-1- and (**d**) VL-COF-2-coated WLEDs in the turn-off and turn-on states. (**e**) CIE coordinates of the commercial blue LED (marked with the solid red triangle), VL-COF-1 (solid red ball), VL-COF-1-coated WLED (solid red ball), VL-COF-2 (red five-pointed star, (0.33, 0.34)), and VL-COF-2-coated WLED (red five-pointed star, (0.27, 0.40)).

## Data Availability

The data presented in this study are available on request from the corresponding author.

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
