# Peer review of "Vinylene-Linked Emissive Covalent Organic Frameworks for White-Light-Emitting Diodes"

_polymers, 2023, doi:10.3390/polym15183704_

Round 1

Reviewer 1 Report

The creation of white light emitting diodes (WOLEDs) constructed by a covalent organic framework (COF) is of great interest. Wan and coworkers developed two vinylene-linked COFs, VL-COF-1 and VL-COF-2, and investigated their PXRD and other properties. Two compounds have been thoroughly characterized and the results are well documented. Therefore, the reviewers consider this manuscript appropriate for publication in Polymers. However, the following minor corrections are required before publication.

Figure 1 “Design and synthesis of VL-COFs”: Looking at the main text and Figure 1, it is unclear why the authors designed VL-COF-1 and -2. The reviewer strongly requests to describe the appropriate molecular design in the main text as well as Figure 1.

Figure 2: Four figures (a)-(d) are too small to see clearly. Resize the figures so that all readers can understand the results.

Regarding the synthetic methods described in Supplementary Information, there is a description of the yield, but there is no information regarding the yield (gram unit) and structure determination of the compound. Add compound states, 1H and 13C NMR spectral data, IR data, high-resolution HRMS data, etc. If solid, the melting point shall also be stated.

The description in English is relatively well written. For further improvement, it is strongly recommended to receive proofreading by an external organization.

Reviewer 2 Report

Wan and coworkers synthesized two emissive COFs, VL-COF-1 and VL-COF-2, and tested their white-light emission performance by coating the COF-dispersed resin epoxy on blue LEDs. The photophysical properties of the two COFs were studied by stead-state and time-resolved spectroscopies. This work provides intriguing candidates for the application of framework-based materials in LEDs. I therefore recommend this manuscript to be published after concerns below being addressed.

1. In Figure 3a and 3b, all absorption and emission peaks were labeled mistakenly. The two peaks at around 400 nm should be the absorption signal while the one at 550 nm should be emission. But both the wavelength and ‘ex’  ‘em’ were labeled totally wrong.  

2.In authors interpretation of FTIR spectra in Figure S1 is incomplete. The intensive absorption of terepthalaldehyde at 1600 cm-1 stems from carbonyl groups. The authors need to compare such absorption with the weak-medium absorption at similar range in COF and explain how the difference demonstrates the connection between the trimethyltriazine and terepthalaldehyde.

3.The authors reported the quantum yield of VL-COF-1 and VL-COF-2 being 12.7% and 11.2%. How good are these results comparing with other emissive COFs?

4. The authors stated, at the bottom of page 4, that ‘Both COFs exhibit very fast fluorescence decay, with fluorescence lifetimes of 10.12 and 7.51 ns, respectively. These results indicate that the fabricated VL-COFs possess excellent emission activity’. It is unreasonable to correlate the fast decay of fluorescence with good emission performance. This sentence needs to be rephrased.

5. The two COFs exhibit different emission lifetime, quantum yield, and white-light emission performance. The authors need to explain what are the reasons, from the aspect of COF structure, to cause their different photophysical properties and what can we learn from the comparison of these two COFs in the design of efficient emitting COFs?
